# Club Cells—A Guardian against Occupational Hazards

**DOI:** 10.3390/biomedicines12010078

**Published:** 2023-12-28

**Authors:** Marina Ruxandra Otelea, Corina Oancea, Daniela Reisz, Monica Adriana Vaida, Andreea Maftei, Florina Georgeta Popescu

**Affiliations:** 1Clinical Department 5, “Carol Davila” University of Medicine and Pharmacy, 050474 Bucharest, Romania; marina.otelea@umfcd.ro; 2Department of Physical Medicine and Rehabilitation, “Carol Davila” University of Medicine and Pharmacy, 050474 Bucharest, Romania; 3Department of Neurology, “Victor Babeș” University of Medicine and Pharmacy, 300041 Timișoara, Romania; reisz_daniela@yahoo.com; 4Department of Anatomy and Embryology, “Victor Babeş” University of Medicine and Pharmacy, 300041 Timișoara, Romania; vaida.monica@umft.ro; 5Doctoral School, “Carol Davila” University of Medicine and Pharmacy, 050474 Bucharest, Romania; andreamutu03@gmail.com; 6Department of Occupational Health, “Victor Babeş” University of Medicine and Pharmacy, 300041 Timișoara, Romania; popescu.florina@umft.ro

**Keywords:** club cell, fibrosis, occupational exposure, CCSP/CC16, biomarkers

## Abstract

Club cells have a distinct role in the epithelial repair and defense mechanisms of the lung. After exposure to environmental pollutants, during chronic exposure, the secretion of club cells secretory protein (CCSP) decreases. Exposure to occupational hazards certainly has a role in a large number of interstitial lung diseases. According to the American Thoracic Society and the European Respiratory Society, around 40% of the all interstitial lung disease is attributed to occupational hazards. Some of them are very well characterized (pneumoconiosis, hypersensitivity pneumonitis), whereas others are consequences of acute exposure (e.g., paraquat) or persistent exposure (e.g., isocyanate). The category of vapors, gases, dusts, and fumes (VGDF) has been proven to produce subclinical modifications. The inflammation and altered repair process resulting from the exposure to occupational respiratory hazards create vicious loops of cooperation between epithelial cells, mesenchymal cells, innate defense mechanisms, and immune cells. The secretions of club cells modulate the communication between macrophages, epithelial cells, and fibroblasts mitigating the inflammation and/or reducing the fibrotic process. In this review, we describe the mechanisms by which club cells contribute to the development of interstitial lung diseases and the potential role for club cells as biomarkers for occupational-related fibrosis.

## 1. Introduction

A decade ago, Celli and Owen announced the arrival of “the time to shine” for club cells and for their secretory molecule, CC16 [1]. This statement was based on the accumulated data from animal experiments which showed a distinct role of these cells in epithelial repair and in the defense mechanisms of the lung. Another important reason to push for the continuation of the investigation of these cells, was the reporting of the secretion of the club cell secretory protein (CCSP) after exposure to environmental pollutants, followed, in chronic exposure, by a reduction in CCSP. The interest in these cells in lung fibrosis is due to their role in the regeneration of injured epithelia, in the metabolism of respiratory toxicants, and due to the secretion of CCSP, which modulates the activity of TGBβ and IFNγ in the distal airways [2].

Exposure to occupational hazards certainly has a role in a large number of interstitial lung diseases. A recent common statement by the American Thoracic Society and European Respiratory Society defined an attributable fraction to almost 40% of the total interstitial lung diseases [3]. Some of them are very well characterized (pneumoconiosis, hypersensitivity pneumonitis); others are consequences of acute exposure (e.g., paraquat) or persistent exposure (e.g., isocyanate). The category of vapors, gazes, dusts and fumes (VGDF), a frequent complex exposure in different branches of industry or in construction, was proven to produce subclinical modifications (high attenuation areas on computer tomography) 2.64% greater (95% confidence interval, 1.23–4.19%) than in non-exposed patients [4]. Hopefully, the involvement of occupational physicians in the multidisciplinary discussion of interstitial lung diseases will increase the currently low percentage of recognized contribution of occupational exposures in idiopathic pulmonary fibrosis (IPF) pathogeny and the misclassification of diagnosis [5].

An acute exposure to high concentrations of respiratory hazards leads to lung injury which might evolve with a normal repair or a fibrotic one. The evolution toward fibrosis depends on the intrinsic properties of the respiratory hazard, the duration of exposure, and the functionality of the defense mechanism, in particular of the anti-inflammatory process which limits the inflammation promoted by the destruction of normal cells.

Chronic exposure to fibers and particles stimulates phagocytosis by pro-inflammatory macrophages. Depending on the dimensions of the particles, their structure, surface reactivity, and chemical transformation, macrophages cannot digest the particles, and a “frustrated phagocytosis” develops. This frustrated phagocytosis is characterized by plasma membrane perturbation and lysosome damage and it triggers chronic inflammation in the lung via Nod-like receptor family pyrin domain containing 3 (NLRP3) inflammasome activation and interleukin-1β (IL-1β) production [6,7]. 

The inflammation and altered repair processes resulting from exposure to occupational respiratory hazards create vicious loops of cooperation between epithelial, mesenchymal, innate defense mechanisms, and immune cells.

The secretions of club cells modulate the communication between macrophages, epithelial cells, and fibroblasts, mitigating the inflammation and/or reducing the fibrotic process. It is the reason why, in this review, we describe mechanisms by which club cells contribute to the development of interstitial lung diseases. Based on the scientific data available, we explored what are the secretory molecules of club cells that could be candidate biomarkers for occupational-related fibrosis.

## 2. General Characteristics of Club Cells

Club cells (formerly named Clara cells) are a population of non-ciliated, non-mucous- secreting cells. Their current name derives from their club shape. They represent 11–22% of cells in the respiratory bronchioles [8]. Club cells are the majority of the secretory cells in the distal regions of the lung [9].

Club cells were first identified by Kolliker in 1881, but their ultrastructure was described almost one century later by Popper and colleagues [10]. Club cells contain a large basally situated nucleus, a small amount of rough and smooth endoplasmic reticulum, but have a large number of mitochondria and electron-dense secretory granules in their apical part. Club cells secrete numerous proteins, the largest amount of which is represented by the secretoglobin family 1A member 1 (Scgb1a1; Gene Cards). Although Scgb1a1 is the official name of this protein on Gene Cards, the scientific literature also uses the name of club cell secretory protein-10 (CC-10). club cell secretory protein-16 (CC-16), club cell secretory protein (CCSP), or uteroglobin. This protein is a homodimer of 70 amino acids, with an exact molecular mass of 15,840 Daltons measured by mass spectrometry [11], from which the name of CC-16 was proposed. The molecular size measured by SDS-PAGE identified a dimension of 10 kDa [12], which led to the abbreviation of CC-10. The difference is not dictated by the structure or function, but the electrophoretic mobility, which influences measurement in the SDS-PAGE technique. Scgb1a1 is expressed by the endometrium (from which derived the uteroglobin name), prostate, thyroid, or mammary gland [13]. The secretion is likely to be much lower than in club cells and/or they bind to the local receptors; therefore, they do not influence the plasma levels of the protein. From the lungs, CC-16 leaks to the general circulation and is rapidly cleared by the kidney proximal tubule cells, having a short plasma halftime. For example, in acute respiratory distress syndrome (ARDS), this halftime is less than 18 min [14]. In addition to CCSP, club cells also secrete other members of the SCGB1A family, SCGB31 and SCGB3A2, which are now well characterized, and which share numerous functions with the CCSP [15]. As there are very few data on their relationship with exposure to respiratory hazards and interstitial lung diseases, this review will focus only on the club cell secretory protein (CCSP), which is an accepted name for the SCGB1A1 gene product.

In healthy non-smoker individuals, the concentration of CCSP in the bronchoalveolar lavage (BALF) is estimated as 1–5 μg/mL. Mean plasma CCSP concentrations in healthy human non-smokers are reported to be around 10–15 ng/mL, but this can be highly variable depending on the assay used [16].

Claudin 10 is a component of tight junctions and has functions in paracellular epithelial permeation [17]. It is considered a marker of early-differentiated club cells and an early sign of the repairing of the bronchiolar epithelium.

Other proteins with potential interest for lung fibrosis are the surfactant proteins A (SP-A) and D (SP-D) [18]. Club cells contribute to the secretion of SP-A and SP-D, but the main producer is type II alveolar epithelial cells. SP-A and SP-D are large proteins, included in the generic term of “collectins”. Collectins are a group of pattern recognition molecules which bind to carbohydrates and/or lipid targets on the surface of various pathogens [19]. In addition to their role in defense against microorganisms, they also participate in the repair process after lung injury.

Cytochrome P-450 (Cyp) family members, namely P450 oxidoreductase, the cytochrome P450, family 2, subfamily F, polypeptide 1 (CYP2F1); the cytochrome P450, family 4, subfamily B, polypeptide 1 (CYP4B1); and the cytochrome P450, family 2, subfamily B, polypeptide 6 (CYP2B6) were identified through transcriptomic analysis [9]. The cytochrome P450 system plays a major role in xenobiotic metabolism, particularly in the phase I processes of oxidation.

## 3. Functions of Club Cells Related to the Pathogenesis of Interstitial Lung Diseases

### 3.1. Progenitor and Repair Function

In their quiescent state, club cells present as differentiated cells; after injury, this status changes into a highly proliferative one. For this reason, they are characterized as facultative progenitor cells [20]. Club cells are not a homogenous population. For example, those involved in the repair process lack cytochrome P-450 enzymes [21]. This is why, some club cells are not as susceptible to naphthalene, for example, and are not destroyed after acute exposure. In the absence of basal cells, club cells retain the ability not only to self-renew but also to differentiate into other types of epithelial cells [22]. So, in the proliferative state, they act as progenitor tissue-specific cell stem cells that retain an undifferentiated character but have a finite capacity for proliferation. The sole function of transit-amplifying cells is the generation of sufficient specialized progeny for tissue maintenance [23]. It is estimated that they represent 15–44% of proliferating cells [24].

The epithelium of the lung is slowly renewed in the steady state, but after injury it rapidly requires the capacity to regenerate. Distal migration of club cells after ozone exposure and bronchiolization of alveoli during lung injury [25] is a demonstration of the involvement of these cells in normal repair. In fact, after three-time administration of bleomycin, there is a significant increase in CCSP+ cells in the lung, probably compensating for the loss in alveolar cell type II [26]. Prolonged or repeated exposure to environmental hazards creates an inflammatory milieu and destroys regenerative cells and the basal membrane, surpassing the capacity of regeneration [27]. The inflammatory cells, orchestrated by macrophages, will eventually lead to the accumulation of fibrocytes, fibroblast transformation and growth, and extracellular matrix deposition, which characterize abnormal tissue repair and are characteristic of lung fibrosis [28]. The disruption of the blood barrier integrity allows SP-A and SP-D, which are hydrophilic proteins with relatively large dimensions, to pass into the circulation. Therefore, they are considered biomarkers of severity in IPF [29].

### 3.2. Metabolism of Xenobiotics

The enzymatic equipment of the lung for metabolizing xenobiotics is far less than that of the liver. However, Hukkanen J, et al. [30] argued for a reconsideration of the role of the lung in xenobiotic metabolism based on the following arguments: that the majority of toxicants are absorbed via the respiratory route; that the lung receives 100% of the cardiac output and the liver only a quarter of it [31]; and that xenobiotics which penetrate the body by any routes other than the digestive system will reach the lung before passing through the hepatic circulation. Examples of toxicants metabolized by club cells are 1-nitronaphthalene or benzo[α]pyrene from cigarette smoke and diesel exhaust; 3-methylindole; 4-ipomeanol; coumarin; di- and trichloroethylene; stiren; furan; and ozone [30,32,33]. At least for some toxic substances (e.g., naphthalene), there are experimental data showing that repeated exposures, despite an increase in dose, reduce the expression of P450 cytochrome and lower the metabolization rate of the toxin [34].

Apart from cytochrome P450 enzymes, club cells also express glutathione S-transferases, dehydrogenases, and aldo-keto reductases [9], which potentially contribute to protection from oxidative stress. In fibroblasts, glutathione S-transferase activates the transcription factor c-Jun [35] and also FAS protein, a proapoptotic cell surface molecule of the TNF family, [36] enhancing the fibrogenic process. We did not retrieve any article on Web of Science or PubMed referring to these molecular events in fibroblasts generated by the release of glutathione S-transferases from club cells, but studies in lungs from COPD patients showed that after induction related to toxicant exposure, glutathione S-transferases can be transported in the extracellular space, as they were present in sputum and in the airways, including the alveolar epithelium and alveolar macrophages [37]. Whether this is of relevance for lung fibrosis remains to be determined. For now, data from studies with various mixtures of pulmonary cells showed higher basal and lower induced oxidative activities in club cells than in alveolar type II cells [38]. This might be explained by the induction, in club cells, of the expression of (Nrf2)-regulated genes by environmental hazards with the potential to generate oxidative stress, such as smoking. Among these Nrf-2 regulated genes are cytochrome P450 1b1, glutathione reductase, thioredoxin reductase, and members of the glutathione S-transferase family, as well as Nrf2 itself [39].

### 3.3. Secretion of Club Cells and Their Relation to the Fibrotic Process

The secretory function of club cells influences the fibrotic process. In CC-16 deficient mice, the club cell secretory granules have ultrastructural changes, and the composition of epithelial lining fluid is altered [40]. This might lead to a less effective defense against environmental factors in the context of decreased CCSP secretion.

CCSP and SP-A have significant roles in modulating both inflammation and the fibrotic process (Figure 1).

#### 3.3.1. Modulation of the Inflammatory Process

Nuclear factor-κB (NF-kB) modulates the innate defense response through a plethora of cytokines and chemokines (IL-1, IL-6, IL-12, IL-18, TNFα, MCP-1, RANTES, CXCL1, CXCL6, etc.); polarization of macrophages to the pro-inflammatory (M1) phenotype; and recruitment of neutrophils and effector inflammatory T lymphocytes [41]. It has been shown that, in a human bronchial epithelial cell line, the induction of CCSP blocked NF-κB activation and the subsequent increase in IL-1β and IL-8. In that experiment, CCSP inhibited the canonical pathway of activation of NF-kB, that is, the inhibition of phosphorylation of the NF-κB inhibitory protein (IκB kinaseα) [42], implicitly avoiding further degradation of this enzyme by the proteasome. The release of IκB kinase α from its binding to NF-kB was stopped by CCSP and transfer of NF-kB proteins in the nucleus was avoided [41]. Similar findings (inhibition of IκB kinase α) were found in mice macrophages in contact with recombinant CCSP and LPS [43]. Furthermore, in that experiment, the recombinant CCSP also inactivated p38 mitogen-activated protein kinase (p38 MAPK). The activation of p38 MAPK also raises the production of pro-inflammatory cytokines (IL-1β, TNF-α, and IL-6) and induces the expression of COX-2, iNOS, VCAM-1, and proliferation and differentiation of immune system cells [44].

Club cell secretions are also involved in the metabolism of arachidonic acid by interfering with phospholipase A2 (PLA2) activity. PLA2 enzymes are a group with 19 components, divided into two major groups, depending on their site of action: secretory PLA2 (sPLA2) and cytosolic PLA2 (c PLA2) [45]. They generate lipids with important functions in the inflammatory process. In particular, in the lung, cPLA2 is requested in the first steps of the arachidonic acid metabolism.

In mice exposed to bleomycin, disruption of PLA2G4A, the gene producing cPLA2IVA, attenuated the overproduction of thromboxane and leukotrienes and the development of lung fibrosis [46]. cPLA2 is also present in the acute lung injury produced by clorhidric acid. The inhibition of cPLA2 reduces pulmonary edema, PMN sequestration and the deterioration of gas exchange [47].

cPLA2 are enzymes involved in the initial steps of arachidonic acid metabolism, which produces pro-inflammatory and pro-fibrogenic molecules, but also anti-inflammatory and antifibrotic mediators, such as prostaglandins E2 (PGE2) and I2 (PGI2). Detailed description of their roles in pulmonary fibrosis have been reviewed elsewhere [48]. In brief, both prostaglandins limit the transformation of myofibroblasts in fibroblasts, the proliferation of fibroblasts, and the synthesis of collagen [49]. The switch from the secretion of protective to deleterious arachidonic derivatives could be the result of epigenetic modifications as shown by an experiment with fibroblasts from IPF lungs compared to fibroblasts from normal lungs [50]. It was also shown that in neutrophils, CCSP inhibits the COX-2 synthase, resulting in significantly reduced levels of PGE2 [51].

In vitro, sCCSP lowered the fibroblastic cytosolic PLA2 activity by 50%. CCSP also inhibited fibroblast chemotaxis in a dose-dependent manner [52]. These in vitro observations suggest a possible antifibrotic role of CCSP, although the complexity of influences of the members of the PLA2 superfamily needs more in-depth analysis for a conclusion.

SP-A suppresses the synthesis of sPLA2-II [53,54] as well as inhibiting hydrolysis of surfactant phospholipids and protecting the lung from external-factor injuries. This is mainly demonstrated for ARDS, but it could be relevant also for less extensive injuries such as prolonged exposure to respiratory irritants.

#### 3.3.2. Modulation of the Fibrotic Process

Transforming growth factor β (TGF-β) is considered the orchestrator of lung fibrosis. TGF-β regulates epithelial–mesenchymal transition, fibroblast proliferation, fibroblast transformation to myofibroblast, and the synthesis of EM [55].

In experimental fibrosis induced by bleomycin, a depletion in club cells was a protective factor for lung injury and against developing fibrosis through a reduction in the expression of TGF-β in the bronchiolar epithelium [56].

More recent data suggest a modulatory effect of CCSP on the fibroblast secretion of TGF-β, with a subsequent reduction of the secretion of the serum amyloid peptide and IL-13 [51], which stimulates fibroblasts and extracellular matrix deposition. There are also some data regarding the pro-fibrotic activity of club cells derived from the observation that in IPF, club cells migrate together to other epithelial lung cells in the alveolar wall and the fibroblastic foci. These club cells have pleomorphic cell shapes and a pattern of various degrees of CCSP or Claudin 10 secretion [17]. In a complex experiment using genetically engineered mouse models and mouse club cell lines, Park et al. [57] demonstrated that Programmed cell death 5 (PDCD5), secreted by club cells, influences the transcription of TGF-β by formation of a Smad3/PDCD5/β-catenin complex and that this interaction promoted EMT and fibrosis. They also showed that PDCD5 was significantly increased in lung samples from IPF patients compared to controls [57].

In cultured macrophages, SP-D, a molecule which is also secreted by club cells, inhibits the secretion of TGF-β and the chemokine migration of fibroblasts in the lung, either directly or through upregulating the platelet derived growth factor AA (PDGF-AA) [18]. After naphthalene exposure, the restoration of club cells in the bronchiolar area is associated with an increase in several growth factors, including TGF-β [58], suggesting a role of TGF-β in club cell regeneration.

Through CCSP, club cells might interfere mucin secretion. In normal human airways, MUC5AC and MUC5B were colocalized with CCSP-positive secretory cells in the proximal superficial epithelia and CCSP+/MUC5B+/MUC5AC− were the dominant cell type in the distal bronchiolar superficial epithelium [59]. In bleomycin-induced pulmonary fibrosis, the concentration of MUC5B in airways is directly related to the severity of pulmonary fibrosis [60]. The over-secretion of MUC5B, genetically determined [61] or induced by exogenous stimuli [62], breaks the mucosal host defense, damages the alveolar type II cells, interferes with the alveolar repair [63], and is associated with an IPF/UIP pattern [64].

These results should be complemented by the finding that CCSP in normal human bronchial epithelium attenuates the response of the secretion of mucin MUC5AC and MUC5B induced by IL-13 [65], a characteristic of club cells which might be lost in ILDs.

The population of club cells in the fibrotic tissue of patients with ILD is different and apparently more heterogeneous compared to that of healthy subjects. Some authors have found less club cells (identified by the SCGB1A1+ marker) in the small airways of ILD, particularly in areas with a UIP radiological pattern [66], and have explained the bronchiolization process by the migration of club cells to more proximal sites in the airways or by the blockade in replacement of these cells after lung injury.

In another study, the dual-labelled (CCSP/tumor necrosis factor-related apoptosis-inducing ligand) club cells were present in the bronchiolar walls of IPF patients and absent in the small airway regions of control lungs [67]. In that study, the interaction between club cells and the tumor necrosis factor-related apoptosis-inducing ligand triggered the apoptosis of the epithelial cells, a phenomenon that is frequently observed on lung biopsies from patients with ILD. It is well known that apoptotic cells and apoptotic bodies initiate efferocytosis, which is sufficient to initiate lung fibrosis [68]. It has also been shown that macrophages participating in the efferocytosis process over-secrete TGF-β [69]. In turn, TGF-β increases the apoptosis of the epithelial cells, perpetuating a vicious cycle [70].

More proof of heterogeneous and dysfunctional club cells in ILDs comes from transcriptomic research. In a comparison of pulmonary cells from IPF and normal donors, SCGB1A1+MUC5B+ club cells were significantly increased in IPF. Secondly, the secretion of the SCGB1A1+SCGB3A2high subpopulation was significantly altered. The SCGB1A1+MUC5B+ club cells from IPF patients expressed mucin (SPDEF, MUC5B, TFF3, and AGR2) and chemoattractant cytokines for immune cells (CXCL1, CXCL6, CXCL8, and CX3CL1). The SCGB1A1+SCGB3A2high subpopulation was not quantitatively different, but had an abnormal transcriptome compared to controls, with high collagen and fibronectin secretion [71].

Epithelial cells also change phenotype in ILD. This has been shown in animal models [72]. Thus, an explanation for the increase in serum and bronchoalveolar lavage of the CCSP is not solely the production of club cells but also a result of the dysfunctional alveolar epithelia. This increase was most prominent in IPF, but present also in other forms of ILD, such as hypersensitivity pneumonitis and connective tissue lung fibrosis [73].

Experimental data are rather contradictory on the collaboration between TNF and club cells in ILDs. On one side, TNF induced CCSP secretion in club cells [74]. This effect was also noticed after exposure of club cells to diesel exhaust, as a useful restoration of CCSP secretion after the DEP-generated methylation of C/EBPα promoter of SCGB1A1 [75].

On the other hand, lipopolysaccharides and TNFα have a synergic effect on the secretion by club cells of keratinocyte-derived chemokine, a cytokine involved in neutrophil recruitment [76]. Neutrophils are necessary for IFNγ and granuloma development, at least in experimentally induced hypersensitivity pneumonitis [77]. IFNγ mediates the recruitment of CXCR3+/CD4+ T cells into the lung through several chemokines (CXCL-9, CXCL-10 and CXCL-11) characteristic of the Th1 pathway [78]. This leads to another interference of club cells with fibrosis, namely the inhibition of the production of IFNγ [79].

IFNγ seems to be necessary for the induction of fibrosis by bleomycin [80,81] but the suppression of IFNγ-related genes was reported in fibroblasts from IPF and for patients with lung fibrosis associated with systemic sclerosis [82].

There are also reports about beneficial, antifibrotic effects on lung fibrosis. For example, in the initial steps of the fibrosis process, cellular-activated pathways dependent on IFNγ inhibit inflammasome formation and inhibit the stability of the integrity phagosomes containing silica, presumably reducing the frustrated phagocytosis [83]. Treatment of IFNγ had a synergic effect on normal and IPF-derived fibroblasts activation and differentiation [84].

## 4. Club Cell Potential Biomarkers in Occupational Interstitial Lung Diseases

In the following sections, we grouped the studies concerning the occupational hazards effects on club cells according to the main categories of occupational fibrosis (https://www.dir.ca.gov/dwc/DWCPropRegs/2020/MTUS-Evidence-Based-Updates-June-2020/Occupational-Interstitial-Lung-Disease.pdf, accessed on 8 November 2023).

### 4.1. Inorganic Particles

#### 4.1.1. Silica

Silicosis, one of the oldest occupational diseases, is still present nowadays. The mixture of new jobs and industries such as artificial stone and sand blaster denim, with the traditional exposures in mining, foundries, sandblasting, ceramics, and glass industries is different between countries. Due to the chronic evolution of silicosis and the carcinogenic potential of free silica, workplace exposure represents an important health issue all around the globe.

The mandatory diagnostic criterion for silicosis is the comparison of the patients’ X-ray with standard radiographic films elaborated by ILO (last revision 2011). The early stages of silicosis are predominantly asymptomatic or mildly symptomatic, without clearly identifiable radiological aspects. As in other interstitial lung diseases, X-ray is not as sensitive as CT for the early detection of the lesions, but CT is not yet standardized for the diagnosis and identification of biomarkers for the early stages of silicosis or for the evolution towards massive progressive fibrosis; it could be an effective solution in the future.

Club cell protein 16 (CC16) appears to be an important marker for the early diagnosis of silicosis. Many studies have shown that the CC16 concentration decreases in silicotic patients. The silica-free dust particles can damage club cells either directly, or can do so indirectly by activating macrophages which release cytotoxic mediators. Concentration and duration of exposure to crystalline silica dust were negatively associated with plasma CC16 levels (*p* < 0.05 and 0.001, respectively), after adjusting for time spent in the community, current smoking, comorbidities, and pulmonary function [85]. This finding is based on an analysis of 57 stonemason workers regularly exposed to crystalline silica and 20 unexposed control workers.

There are also studies that tried to identify the cut-off value for the early stages of silicosis. Nandi et al. [86] measured serum CC16 levels in 106 subjects (68 silica-exposed and 38 healthy individuals). By ELISA measurement of CC16, all subjects radiologically confirmed with silicosis had serum levels below 9 ng/mL, while healthy subjects showed CC16 > 9 ng/mL. Therefore, CC16 was proposed as a possible screening tool for early detection of silicosis among workers with a history of silica dust exposure. The recommendation was emphasized due to the relatively low cost and because the screening can be used in disadvantaged communities. The same cut-off was confirmed in another investigation of 117 silicosis subjects vs. 32 unexposed individuals [87]. The degree of pulmonary damage in this study was assessed with the Lung Damage Score: (LDS) = (X × Y × Z) + L, a formula in which X referred to small opacities, Y to the profusion of small opacities, Z to the number of affected lung zones, and L to the score of large opacity, based on the ILO classification. The age and the duration of exposure were also considered when designating the cut-off. Another research group [88] found another cut-off (16.21 ng/mL), with an 81.10% sensitivity and 92% specificity. The study included 239 men, 75 with silicosis, 75 exposed to crystalline SiO_2_ without X-ray modifications suggestive of silicosis, and a control group of 89 ealthy, non-exposeh d persons. This cut-off also considered age, duration of exposure, smoking, and alcohol consumption. There was no distinction between the stages of silicosis, although the authors mentioned that the silicosis group included patients in all three stages of the disease. This differentiation was investigated in another pilot study with 121 participants by Naha et al. [89]. They found that in healthy non-smokers, cut-off values of 13.0 and 7.0 ng/mL could be considered to distinguish early-stage (asymptomatic phase) silicosis from terminal or advanced silicosis. For smokers, values of 9.0 and 5.0 ng/mL confirmed the additive effect of smoking. Sensitivity and specificity were high enough (≥83%) to recommend this biomarker for screening purposes. These results are promising but, as the authors mentioned, need to be validated on a larger population, analyzing different stages of the disease, for a wider range of exposures and industries and from different geographic areas.

CC16 was also determined in the BALF retrieved from the lungs of patients in different stages of silicosis [90]. The BALF CC16 was positively correlated with FEV1/FVC and VC max. Unfortunately, lung function deterioration is not specific to silicosis. The CC16 levels in BALF from patients with silicosis stage I and II were smaller, compared to the control group of exposed workers with respiratory symptoms but without radiological confirmation of silicosis. With the progression of silicosis, the CC16 level in BALF increased, regaining (in stage III) the level of the control group. These conflicting results might be related to the bias in the selection of the control group and to the very low number of stage III silicosis patients (13). As CC16 is also a potential biomarker for chronic obstructive pulmonary disease, its diagnostic value might be reduced for silica-exposed workers who might suffer from occupational bronchitis.

Experimental data complement these clinical data: a recent RCT [91] on a rat model showed that the serum level of CC16 was negatively related to lung injury and silicosis. Fibrosis models were constructed, and rats were randomly divided into control and silica-exposed groups. CC16 was measured in BALF supernatants and serum using an ELISA kit. CC16 concentrations decreased in both BALF and serum after silica exposure, and the reduction increased with silica dose in BALF and serum. Regression analysis showed that there was a significant and negative correlation between BALF CC16 concentration and silica content. This study had the advantages of studying fibrosis in vivo, of using randomization to reduced biases, and of a rigorous tool to examine the cause–effect relationship between lung injury and serum CC16 level. However, a direct transition to humans is not possible.

In summary, most evidence points to a significant reduction in serum CC16 levels in workers chronically exposed to silica dust, without respiratory symptoms and with normal chest x-rays and normal pulmonary function tests. These results recommend serum CC16 as a possible tool for detecting early asymptomatic silicosis in the silica-exposed population. More evidence is needed because in most studies either the number of patients was small, or data was not sufficient to determine an agreement on a relevant cut-off value.

#### 4.1.2. Asbestos

Asbestos has been known since antiquity and used particularly for its insulating properties: thermic, electrical, sound-proof and fireproof. It is present with numerous combinations of hydrated silica of Mg, Fe, Ca and Na. The most used forms are chrysotile, crocidolite, amosite, anthophyllite, tremolite, and actinolite.

Exposure to asbestos fibers can affect both the lung tissue and the pleura, developing in time into lung asbestosis, pleural effusions, pleural plaques, pachypleuritis, lung cancer, or malignant mesothelioma.

The lesions resulting from the asbestos exposure are located at the bifurcation of the terminal bronchioles into alveolar ducts. The injury of the alveolar wall epithelium is due to the incomplete interstitial phagocytosis of the fibers by the macrophages, and the release of pro-inflammatory factors, cytokines (TNF-alfa, IL-1, IL-6), growth factors or free radicals. The inflammation stimulates fibroblasts proliferation and the deposit of collagen fibers.

To assess the contribution of club cells in asbestos-related lung reaction, CC-16 in serum and several proteins in BALF (CC16, SP-A and PLA2) were compared between 34 workers exposed to asbestos and 39 non-exposed individuals [92]. Both BALF CC-16 and SP-A were increased in the group exposed to asbestos. This increase appeared early, without statistical differences between CC-16 and SP-A, according to the existence or non-existence of the radiological findings in smokers. In exposed workers without signs of radiological impairment, the CC16 in BALF was 2.255 ± 588 microg/L, similar to the one in already diagnosed asbestosis. Smoking increased the influence of the CC-16 level, whereas SP-A was not significantly modified. The CC-16 serum/BALF ratio showed a considerable increase in the smoker group but was less marked in those exposed to asbestos or with asbestosis. PLA2 activity in BALF was slightly increased in the study group compared to the control one, without being influenced by the stage of the disease or by smoking. Age did not significantly influence CC-16.

Another cross-sectional study analyzed the level of CC-16 in both serum and BALF; the albumin level; and CD4/CD8+ and T lymphocytes in BALF in relation to the exposure time and the pulmonary deterioration stage [93]. These parameters were consecutively monitored both in a group of 34 workers exposed to asbestos and in a similar size control group without exposure. The exposure time was divided into short (<15 years) and long (over 15 years). The severity of the lesions was classified into three categories: without lesions, with pleural disease deterioration, and with pulmonary parenchyma deterioration. The results of this research showed a significantly high level of CC-16 in the serum and BALF of the exposed patients, but showed lower values in the smokers’ group. The serum CC-16 values positively correlated with the BALF values. The serum concentration of CC16 was significantly increased in asbestos-exposed subjects, reaching values of 27.2 ± 24.0 mg/L in asbestos-exposed subjects (*n* = 34) compared to 16.1 ± 7.6 mg/L in the control group (*n* = 34); *p* = 0.01. Smoking played an important role, with significant differences between serum CC16 and BALF levels, generally lower in smokers (*p* = 0.05 and *p* = 0.001, respectively) compared to higher levels in non-smokers.

No link was found between the exposure time and the degree of pulmonary deterioration, on one side, and the CC-16 in serum and albumin, CD4/CD8 and T lymphocytes in BALF, on the other. In conclusion, CC-16 increased after a short-term exposure, preceding the radiological modifications, but remained unaltered throughout the disease evolution.

There are also some experimental studies supporting a role for club cells in asbestosis. Two experimental studies on rats [94,95] studied inflammatory and proliferative changes and other markers of fibrogenesis in asbestosis. One of these experimental studies investigated the effect of potassium octatitanate whiskers (PT1), an asbestos substitute, on male Wistar rats [94]. After 3 days, as well as after 3 and 6 months from the administration of a single intratracheal instillation of 10 mL PT-1, CCSP and TTF-1 (Thyroid Transcription Factor-1) mRNA were reduced. The level of SP-A mRNA also decreased after 1, 3, and 6 months. This experiment showed that CCSP and SP-A are involved in both the acute and chronic response to exposure to PT1. Thus, the decrease of the factors that normally inhibit the fibrotic process in the lungs will lead to the development of pulmonary fibrosis.

The MEK1 (mitogen–activated protein kinase kinase-1) role of signaling epithelial cell replication and lung remodeling after asbestos injury was demonstrated by asbestos-exposed transgenic mice, expressing a dominant-negative MEK1 (Tg+) vs. transgenic–negative mice (Tg−), exposed to crocidolite asbestos. After 32 days of exposure, in Tg+ mice, the club (Clara) cells differentiated into ciliated and mucin-producing cells and decreased CCSP mRNA levels [95].

More detailed investigation of the relationship between asbestos exposure and CC16 levels remains to be addressed by new studies. It is not possible to propose from the present data reference values for serum CC16 levels in asbestosis.

#### 4.1.3. Nanomaterials

The health impacts observed in animal studies focusing on the inhalation of engineered nanoparticles encompass pulmonary fibrosis, granuloma formation, inflammation, lung cancer, mesothelioma-like effects, cardiovascular consequences, oxidative stress, and the formation of pleural plaques [96,97]. In addition, needle-like fibrous carbon nanotubes elicited asbestos-like granuloma formation and an increased risk of mesothelioma in a mouse strain predisposed to tumors (NIOSH, 2013).

Evidence of human health effects of ultrafine particles, (e.g., lung inflammation, oxidative damage, exacerbation of heart disease, atherosclerosis, asthma, and potentially even lung cancer), was gathered from epidemiological studies. These studies primarily focused on inadvertently produced nanoparticles originating from sources like traffic pollution and combustion by-products such as diesel exhaust and welding fumes [98].

A longitudinal study [99] conducted in 14 manufacturing plants in Taiwan enrolled 124 nanomaterial-handling workers and 77 non-exposed controls. The nanomaterials handled by the group of exposed workers were carbon nanotubes, silicon dioxide, titanium dioxide, and other nanomaterials, including nanoresins, nanogold, nanosilver, nanoclay, nanoalumina, and metal oxides. At 6 months of follow-up, the lung damage markers, especially CC16, were associated with the group of workers handling nanomaterials. In the exposed group, the reduction of the serum CC16 between the baseline and the 6-month follow-up value was significantly greater compared with the control group.

#### 4.1.4. Coal

Exposure to coal dust can lead to coal worker pneumoconiosis (CWP). A Chinese study assessed the utility of certain blood biochemical markers, among which was CC16, level [100] for monitoring coal dust-induced early lung injuries and the stage of coal worker’s pneumoconiosis (CWP). There were two groups consisting of, first, tunneling workers (34), coal hewers (13), and ancillary workers (17 were considered as in the control group due to their very low exposure), and the second group consisted of patients with CWP in different stages (45). The results showed that there was no difference in CC16 serum level within the workers in the first group (x2 = 2.94, ν = 3, *p* = 0.40), nor between the workers in the first group and the CWP patients in the second group (*p* = 0.20, ν = 2, *p* = 0.90). After analyzing the independent variable of smoking, alcohol consumption, job, age, job seniority, and CWP stage with serum CC16 level, the most relevant conclusion was that there was a strong association between the increase in job seniority and the decrease of CC16 serum level (*p* < 0.05, OR = 0.900, 95% CI = 0.823–0.985). Therefore, CC16 cannot be used as a biomarker for early detection and stage of CWP, further research still being required.

### 4.2. Organic Particles

Hypersensitivity pneumonitis (HP) is an interstitial lung disease that involves a complex immunological reaction that appears in the context of occupational or non-occupational exposure to organic dusts or to chemical substances with small molecular mass.

In a relatively recent American study [101] regarding occupational or non-occupational exposure in an HP group of patients, they conducted patient interviews investigating lung functional capacity, antibodies, exposures, HSCRP, CC16, and SST2. Analyzing the inflammatory or fibrotic biomarkers, they observed a high statistical prevalence of HSCRP, CC16, and SST2 when compared to references. For the multivariate analysis, only CC16 was significantly associated with the increased odds of HP, which might recommend CC16 as a valid biomarker, but further investigations would be necessary.

#### 4.2.1. Bioaerosols

Bioaerosols are airborne particles from biological sources, including pollen, bacteria, fungal spores, and viruses and their metabolites and toxins [102]. These agents are released into the air through the handling process of the wastes. During cell lysis or active Gram-negative bacteria cell growth, endotoxins are emitted [103]. Chronic exposure to endotoxins may lead to non-specific inflammation of the airways (organic dust toxic syndrome (ODTS), mycotoxicosis, grain fever, toxic alveolitis) and subsequently to a decrease in lung function by releasing pro-inflammatory cytokines, thus affecting the permeability of the bronchoalveolar barrier.

Inhalation of bioaerosols may lead to irritation of the skin and mucous membranes, toxic effects such as toxic syndrome of organic dust, decreased lung function, gastrointestinal symptoms, and, less commonly, infections and allergies [104,105].

In the baseline examination of a prospective cohort study, Steiner et al. (2005) [106] reported an increased CC16 concentration in non-smoked wastewater workers 11.0 (5.6–23.0) and garbage workers 11.3 (5.6–21.0) vs. a control group 9.4 (4.3–18); *p* = 0.01. The increased CC16 concentration was assumed to be compatible with the hypothesis that bioaerosols cause subclinical alveolitis. Consistent with other studies, the effect of smoking was consistently highly significant and reduced CC16 concentrations. Wastewater exposure and duration of wastewater exposure resulted in increased CC16 concentrations compared to no significant effect in the control group (waste collector or farmer exposure). However, a low adjusted r2 (maximum 6%) indicates a model with low predictive value. These results need to be further verified. In conclusion, CC16 was increased in workers exposed to bioaerosols especially in smokers, suggesting that the serum concentration might be predictive of the risk of later damage to the lung and its function.

#### 4.2.2. Organic Dust

Organic dust is found in agricultural settings and has a heterogeneous composition containing mainly organic particles of plant, animal, and microbial origin [107]. Organic dust particles found in animal confinement buildings consist of animal dander, urine, and feces. These microorganisms, allergens, and toxins are widespread in the air and represent an important health risk to livestock farmers.

The organic fragments of these dusts include G-positive and G-negative bacteria, molds and yeasts, histamine, endotoxins, and even pharmaceutical compounds [108]. Another potential health hazards is noxious gases (e.g., ammonia and hydrogen sulfide) which come primarily from animal droppings and from manure pits. Chronic exposure to high concentrations of organic dusts may lead to chronic bronchitis, COPD, and altered lung function [109,110]; a significant association was found between asthma and organic dust syndrome, hypersensitivity pneumonitis, or animal farming.

An experimental study found a decreased number of club (Clara) cells and altered CC10 secretion in mice exposed multiple times to chicken barn air, compared to the control mice. The longer the exposure, the more significant the decrease in the number of club cells. It was also observed that the number of mucus-producing goblet cells increased post-exposure, probably as a response to the lung injury that can cause proliferation of these airway epithelial cells [111].

Although exposure to organic dust is recognized for all agricultural workers, especially for those involved in raising animals, at present there is little evidence of the role of club cells in the etiopathogenesis and early diagnosis of extrinsic allergic alveolitis.

### 4.3. Mixture Exposure (Vapors, Gazes, Dust and Fumes VGDF)

#### 4.3.1. Smoke Exposure during Firefighting

Smoke inhalation is a major occupational hazard for firefighting personnel. The results of several studies of municipal firefighters suggest that there may be a cumulative effect from repeated exposure to smoke causing chronic pulmonary dysfunction [112]. The smoke contains various low molecular weight oxygenated compounds such as acrolein, nitrogen and sulfur dioxides, aldehydes, and halogenated hydrocarbons which result from the combustion and pyrolysis [113].

A study [114] was carried out on a group of voluntary firefighters who were exposed to smoke from the combustion of polypropylene for about 20 min, in addition to a control group. The concentration of serum CC16 found in firefighters immediately after the exposure was significantly higher than that of controls examined simultaneously. Ten days later, the CC16 serum levels of all firefighters had returned to normal. This study showed a transient increase of club (Clara) cell secretion after a short exposure to smoke. The increase in CC16 serum levels suggests a disruption of the bronchoalveolar/capillary barrier caused by acute inflammatory changes in the airways. Therefore, CC16 might be considered as a biomarker for early detection of the changes in permeability of the bronchoalveolar/capillary barrier resulting from smoke inhalation or other lung irritants.

#### 4.3.2. Coke Ovens

During the coking process, PAH (polycyclic aromatic hydrocarbons) are produced in large quantities [115]. Coke oven workers are predominantly male and smokers. When stimulated by exogenous substances, the airways locally produce large amounts of reactive oxygen species (ROS) and inflammatory factors. On short-time exposure, the club cell secretion of CC16 counteracts these pathological mechanisms [116] but the long-term exposure may lead to a decrease in the number of club cells and a decrease in their CC16 output.

A Chinese longitudinal study [117], developed between 2014 and 2019, enrolled 313 workers from coking plant and investigated the associations between PAH (polycyclic aromatic hydrocarbons), tobacco smoke exposure, and the levels of CC16. They measured urinary nicotine, PAH metabolite levels, and CC16 serum levels. A correlation between exposure to tobacco smoke, low plasma CC16 levels, and a decline in lung function among coke oven workers was found. The reduction in plasma CC16 levels occurred prior to the deterioration in lung function. The coke oven workers with lower plasma CC16 levels were at an increased risk of experiencing a decline in lung function following exposure to tobacco smoke.

#### 4.3.3. Sulfur Dioxide in a Non-Ferrous Smelter

Sulfur dioxide (SO_2_) is a toxic gas released primarily during the burning of fossil fuels, and the roasting of sulfide ores in non-ferrous smelters. The effects of SO_2_, which rapidly occur, consist of irritation symptoms of the respiratory tract, decreased lung function, and mucociliary clearance [118]. These symptoms are shown to be aggravated by exercise, oral breathing, or co-exposure to other pollutants such as particulate matter [119].

The effects of SO_2_ on lung epithelium were studied [120] in a group of healthy male workers exposed to SO_2_ in the non-ferrous smelter and compared to a reference group. SO_2_ exposure was associated with a highly significant reduction in serum CC16 concentration and an increase in serum SP-D, with the CC16/SP-D ratio significantly reduced by an average of 42% in the most exposed workers. The changes in the serum levels of CC16 and SP-D are most likely due to the damage to the respiratory epithelium caused by exposure to SO_2_ and probably some other toxicants present in the atmosphere of the smelter.

#### 4.3.4. Ozone Exposure

Ozone, an environmental air pollutant, is a highly reactive oxidant capable of causing oxidation and peroxidation of membrane lipids and proteins.

The inflammatory/antioxidant responses in the respiratory epithelium to ozone were investigated in rats exposed for 3 hours to ozone [121]. Histological findings of the lung showed significant desquamation and hypercellularity of the bronchiolar epithelium, loss of cilia, necrotic debris in the lumen, perivascular edema, vascular congestion, and a decrease in CC 16 in bronchoalveolar lavage, which was maximal 24 hours post-exposure. Similar changes have been described in the terminal bronchioles of mice, 48 hours after ozone exposure in other experiments [122]. Histological and structural alterations persisted for at least 72 hours post-exposure, suggesting a mechanism of underlying prolonged alterations in the respiratory tract, despite the resolution of inflammation and injury in the lower lung.

#### 4.3.5. Hydrogen Peroxide

Hydrogen peroxide is a compound mainly used in chemical industrial processes, for its bleaching properties (in the paper industry, for bleaching textiles and cosmetics), hair, and as a disinfectant and for water treatment. Hydrogen peroxide is well known to be irritating to the airway mucosa. In order to avoid the potential risk of occupational exposure, closed–automated production systems are often used. Occupational hazards such as spills and leaks are associated with manual handling of hydrogen peroxide in old and/or small industrial settings [123].

The acute effects of hydrogen peroxide on the serum CC16 level, as a marker for the inflammation induced by this irritant, were investigated by Ernstgård L at al. [124]. Eleven healthy volunteers were exposed to two different concentrations of hydrogen peroxide in a controlled exposure chamber. After 2 hours of inhalation, the level of CC16 in serum had a tendency to decrease at both post-exposure time points, the effect being more pronounced at the higher concentration of hydrogen peroxide. Moreover, men tended to have slightly higher values than women.

#### 4.3.6. Diesel Exhaust

Diesel engine exhausts (DEE), a complex mixture of gas and diesel exhaust particles (DEP), are a primary contributor to air pollution around the world. DEP consists of mainly PAHs (polycyclic aromatic hydrocarbons) [125]. According to Pronk et al. [126] occupational exposure to DEP is high for underground mining and construction, medium for working in semi-closed spaces on the ground surface, and low for those working outside or not directly exposed.

Exposure to diesel exhaust has been shown to induce local inflammatory changes and induce T-helper type 2 (Th2) responses [127].

In a double-blind, randomized crossover study [128], atopic individuals were exposed to DEP, and blood samples were obtained 48 hours after exposures and assayed for CC16. As a result of DEP exposure, CC16 decreased. The mechanism by which DEP reduces airways CC16 is unknown, but it is well known that CC16 plays an important role in protecting respiratory tract epithelium against oxidative damage, and that it modulates immune responses to inhaled irritants.

#### 4.3.7. Welding Dusts and Fumes

It is well known that welding fumes, a major issue in occupational medicine, predispose workers to high health risks of respiratory, cardiovascular, and neurological effects [129,130,131]. In particular, welding fumes of stainless steel contain in chromium (Cr) and nickel (Ni), both metals classified by IARC as carcinogenic agents in humans.

An experimental study [132] investigated the effects after 5 days and 10 days of inhalation of stainless steel welding dust particles on CC16 in rats. The BALF of rats showed decreasing club cell levels with exposure to the welding dust. Histological findings showed lung deterioration, which was associated with a significant decrease in CC16 concentration immediately after the instillation of the welding dust samples.

### 4.4. Chemicals

#### 4.4.1. Isocyanates

Isocyanates are characterized by the presence of highly reactive N=C=O groups. These compounds find application in the manufacturing of polyurethane polymers, which are utilized in a diverse range of applications, such as the production of flexible and rigid foams, surface coatings, and adhesives. Spray painters are at risk for developing such symptoms due to potentially high isocyanate exposure, as most lacquers contain hexamethylene diisocyanate. The most frequently used isocyanates are toluene diisocyanate (TDI), diphenylmethane diisocyanate (MDI), hexamethylene diisocyanate (HDI), and biuret-modified HDI (HDI-BT) [133].

Acute toxicity can cause throat irritation with coughing and difficulty in breathing. If the exposure is more severe, it may result in hypersensitivity pneumonitis or even pulmonary edema. Chronic inhalation can lead to immune disorders as well as respiratory tract lesions. [134]. The skin may also be affected [135].

A study investigated 50 workers exposed to isocyanates during car spray-painting and compared their CC16 level to 30 control subjects (smokers and non-smokers never exposed to isocyanates) [136]. Compared to the control group, the exposed workers showed significant lowering in serum CC16 levels. Moreover, the serum CC16 was much lower compared to the non-smokers.

#### 4.4.2. Pesticides

Pesticides have numerous negative health effects on the central and peripheral nervous system, respiratory tract, skin, gastrointestinal tract and reproductive system. [137]. They include a variety of substances, generally classified according to their direct effects on plants or on some vectors of disease for humans (e.g., insecticides, herbicides, fungicides, etc.) [138].

An experiment investigated the behavior of rat club cells following acute exposure to a commercial insecticide or following its repetitive inhalation for 5 days [139]. The intracellular content of CC10 in the respiratory tract was assessed. After a single exposure to insecticide, club cells showed a great expansion in their volume and number and the bronchiolar CC10 content increased. Through repeated exposure to the insecticide, significant alterations occurred in the bronchiolar epithelium of rats, with a particular emphasis on club cells; their numbers and size substantially diminished and the secretion of CC10 was blocked. 

The summary of all the data presented above is shown in the table below (Table 1):

## 5. Study Limitations

Our review highlights several limitations to the use of CC-16 as a biomarker for occupational pulmonary fibrosis. Currently, there are only a few studies conducted on a limited number of participants, which reduces the reliability of the results. The different methods do not allow a clear comparison of the results, which would lead to common conclusions. The few animal studies, although consisting of randomized clinical trials, do not allow the data as such to be transferred to humans.

## 6. Future Perspectives

Occupational medicine has an important preventive component which includes the individual characterization of risk and the early detection of disease. An element shared by many forms of occupational fibrosis is that, in chronic exposure, the evolution is long and depends on the duration and intensity of exposure. Therefore, cessation of the exposure in the early stages of fibrosis could assure a good quality of life for many years. But this intervention has to be based on strong arguments in order to be accepted by the employee, who, in many cases, has to completely change their job. For this reason, the occupational physician needs a reliable biomarker of the early process in order to monitor the exposed workers and to detect the modifications in an early stage.

Due to the experimental data and the current knowledge of club cells, it is reasonable to believe that we might be in front of this potential biomarker. One way of validation is to perform multicentric studies on workers exposed to a common occupational hazard with fibrogenic potential. As cumulative exposure is a key factor, a similar evaluation of the past and current exposure is mandatory for comparable results.

There is also another possible approach, as workers exposed to occupational hazards could benefit from research on club cells focused on other diseases associated with lung fibrosis. If club cell products become reliable biomarkers of fibrosis, knowing that exposure to occupational dust is not only a cause but also an aggravating factor [140], a part of individual risk assessment might include the monitoring of CC16 level in order to prevent the progression of fibrosis.

In view of all mentioned above, research on club cells should be extended to reach a significant exposed population to confirm their potential as early biomarkers of fibrosis.

## 7. Conclusions

Despite extensive literature about club cells there are many unsolved issues and roles to be clarified.

The best-studied molecule produced by club cells in occupational exposure to toxic substances is CC16. From the current available data, we can conclude that CC16 has great potential to become a sensitive biomarker for the negative lung effects of a variety of occupational exposures, from dusts and fibers to chemicals.

The most probable exposure effect is the reduction of CC16 in serum or BALF levels. However, all the studies supporting this finding included relatively small numbers of p and used different techniques of measurement which do not allow the aggregation of the results.

Surfactant proteins derived from club cells were also studied, but in much too low proportions to support any sort of conclusion.

It is worth continuing the investigation of club cell products in occupational medicine, in order to make conclusions on their relevance for monitoring and early detection of the negative health effects related to specific exposures.

## Figures and Tables

**Figure 1 biomedicines-12-00078-f001:**
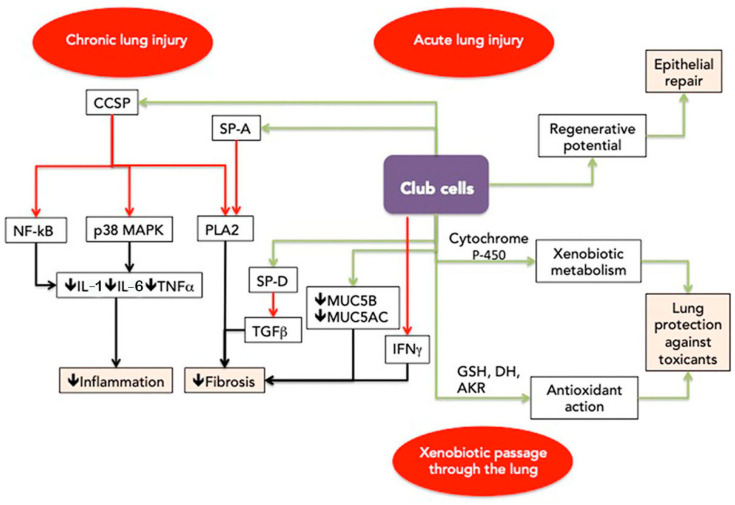
Putative defense mechanisms modulated by club cells. Functional club cells participate in the repair process after acute lung injury, modulate inflammation and fibrosis in chronic interstitial lung diseases, metabolize toxicants, and act against the oxidative stress induced by toxicants. Legend: 
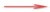
 inhibition; 
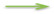
 activation. AKR—aldo-keto reductase; DH—dehydrogenase; CCSP—club cell secretory protein; GSH—glutathione S-transferase; IFNγ—interferon γ; IL-1—interleukin 1; IL-6—interleukin 6; MUC5AC—mucin 5AC; MUC5B—mucin 5B; NF-KB—nuclear factor kappa B; p38 MAPK—p38 mitogen-activated protein kinase; PLA2—phospholipase 2; SP-A—surfactant protein A; SP-D—surfactant protein D; TGFβ—transforming growth factor β; TNFα—tumor necrosis factor α; 🡣 decrease.

**Table 1 biomedicines-12-00078-t001:** Club cells derived biomarkers in occupational lung fibrosis.

Type of Hazard	Hazard	Interstitial Lung Disorder	Club Cells Biomarker	Reference
Inorganic Particles	Free silica	Silicosis	Serum CC16 🡫	[89]
			Serum CC16 🡫	[86]
			Serum CC16 🡫	[87]
			CC16 in BALF 🡫	[90]
			Serum CC16 🡫	[85]
			Serum CC16 🡫	[88]
			Serum and BALF CC16 🡫 SP-D *	[91]
	Asbestos	Asbestosis	Serum and BALF CC 16 🡩 SP-A- BALF ~	[92]
			CC 16 serum and BALF	[93]
			CCSP mRNA * 🡫	[95]
			CCSP mRNA * 🡫	[94]
	Nanotubes and Nanoparticles	Lung fibrosis	Serum CC16 🡫	[99]
	Carbonauceous	Coal worker pneumoconiosis	Serum CC16 🡫	[100]
Organic particles	Bioaerosol	Hypersensivity pneumonitis	Serum CC16 🡩	[106]
	Organic dust	Organic dust syndrome	Club cell * 🡫 CC10 🡫	[111]
Mixture	Vapours, gazes, dust and fumes (VGDF)	Lung fibrosis		
	Smoke		Serum CC16 🡩 transient	[114]
	Polycyclic aromatic hydrocarbons (PAH)		Serum CC16 🡫	[117]
	Sulphure dioxide (SO_2_)		Serum CC16 🡫 Serum SP-D	[120]
	Diesel engine exhaust		Serum CC16 🡫	[128]
	Hydrogen peroxide		Serum CC16 🡫	[124]
	Welding dust		BALF CC16 * 🡫	[132]
	Ozone		BALF CC16 * 🡫	[121]
Chemicals	Isocyanates	Hypersensivity pneumonitis	Serum CC16 🡫	[136]
	Pesticides	Lung fibrosis	BALF CC10 * 🡫	[139]

* experimental models; 🡩 increase; 🡫 decrease.

## Data Availability

Not applicable.

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
