# Peer review of "Club Cells—A Guardian against Occupational Hazards"

_biomedicines, 2023, doi:10.3390/biomedicines12010078_

Round 1

Reviewer 1 Report

Comments and Suggestions for Authors

1.This research focused on  Club cells – a guardian to occupational hazards , after check the pubmed, there were not so many articles aboult this topic, and so this manuscript was very prospective and significant.

2.This manuscript foucus on clinical problems of  lung disease, with strong clinical value and importantce,and also met the submission topic of this journal, but some places can be more perfect.

3.Figure 1 if use professional software to draw such as AI maybe much more better.

4. If there were some clinical trials  for Club cells, please add.

5.English should be more polish.

Comments on the Quality of English Language

Neally OK.

Author Response

Dear Reviewer, 

Thank you for your valuable comments and suggestions regarding our manuscript. We hope to have answered all your guidelines.

Best regards, 

Authors

Reviewer 2 Report

Comments and Suggestions for Authors

In this review, the authors tried to present a comprehensive overview about club cells in occupational hazards. Although the manuscript contains valuable information. However, the writing quality of this review have to substantially improved. The following are some comments:

Line 21-22 “The American ………” incomplete sentence

Line 25 and 52: The abbreviation VGFF is wrong, it should be VGDF; replacing fumes by dust.

Line 42-45:  In what concerns interstitial fibro- 42sis, the interest for these cells was…….” Need to check for appropriate meaning

Lines 47-51 are similar to those in Line 21-24; should be rephrased

Line 53, 57, 69, 104, and 111: All abbreviations should be defined at their first presentation

Line 82-83: “They represent 11-22% of the cells in the respiratory bronchioles were they predominate compared to the ciliated 83cells [8]” please revise for correct meaning

Line 182: These is repeated twice

Line 336: IFN please correct

Line 506: “The assess the contribution of club cells in asbestos related lung reaction, [92], CC-16 506in serum and several proteins in BALF…” please revise for correct meaning.

Line 541: “One of these studies experimental study 541investigated the effect of the potassium….”, Please correct

A separate section including future perspectives should be added.

A new section of Limitations of using CC-16 as a biomarker for occupational hazards should be added. 

Comments on the Quality of English Language

Extensive editing of English language required

Author Response

Dear Reviewer, 

We would like to thank you for all your valuable comments and suggestions concerning our manuscript.

We hope that we have addressed all your requests in view of publication of our article.

Best regards, 

Authors

Round 2

Reviewer 2 Report

Comments and Suggestions for Authors

The authors have responded to all reviewer comments

Comments on the Quality of English Language

Minor editing of English language required